# Comparative Characterization of Key Volatile Compounds in Slow- and Fast-Growing Duck Raw Meat Based on Widely Targeted Metabolomics

**DOI:** 10.3390/foods11243975

**Published:** 2022-12-08

**Authors:** Kaiqi Weng, Lina Song, Qiang Bao, Zhengfeng Cao, Yu Zhang, Yang Zhang, Guohong Chen, Qi Xu

**Affiliations:** 1Key Laboratory for Evaluation and Utilization of Poultry Genetic Resources of Ministry of Agriculture and Rural Affairs, Yangzhou University, Yangzhou 225009, China; 2Joint International Research Laboratory of Agriculture and Agri-Product Safety, The Ministry of Education of China, Yangzhou University, Yangzhou 225009, China

**Keywords:** duck, aroma, electric nose (E-nose), solid-phase microextraction-gas chromatography-mass spectrometry (SPME-GC-MS), metabolomics

## Abstract

The volatile aroma compounds in raw duck meat strongly affect consumers’ purchase decisions and they vary among breeds with different growth rates. In this study, slow-growing (SG) Liancheng White and fast-growing (FG) Cherry Valley ducks were selected, and their volatile compounds were characterized using electric nose and gas chromatography-mass spectrometry. Furthermore, a widely targeted metabolomics approach was used to investigate the metabolites associated with volatile compounds. The results showed that hexanal, nonanal, octanal, heptanal, and 2-pentylfuran were abundantly present in duck meat, regardless of the breed. The higher nonanal and octanal rates contributed to the fatty and fruity aroma in SG meat than FG meat, while FG meat had a mushroom note resulting from higher octenol. Furthermore, widely targeted metabolomics showed a lower carnitine content in SG meat, which might promote lipid deposition to produce more octanal and nonanal. Higher sugar and amino acid contents led to a meaty aroma, whereas more trimethylamine N-oxide may generate a fishy note in SG meat. Taken together, this study characterized the raw duck meat aroma and provided the basic mechanism of the formation of the key volatile compound.

## 1. Introduction

Duck is the major source of meat products in China, with a total production of approximately 3.43 billion tons in 2020, accounting for 68.66% of global duck production [1]. Duck meat is increasingly popular owing to its high nutritional value, as it is a good source of essential amino acids and polyunsaturated fatty acids [2]. In addition to the nutritional value, the flavor is an important factor affecting consumers’ purchasing behavior in the poultry market [3]. The flavor of meat comprises of its taste and aroma, and several studies have characterized the key flavor compounds in cooked duck meat. The main flavor of roasted duck is the strong aroma of fat, roast, and meat, which have been identified to be hexanal, dimethyl trisulfide, 2-furfurylthiol, and others [4]. The flavor compounds of water-boiled salted ducks are mainly furans, ketones, alcohols, aldehydes, and esters [5]. However, these studies have mainly focused on cooked samples, and less research has been conducted on raw duck meat. Although the aroma of raw meat is slight, some compounds that are present in raw meat will remain after cooking and should be acceptable by consumers before purchasing [6]. Thus, more effort is needed to characterize the key aromatic compounds in raw duck meat.

Additionally, similar to chickens, duck meat is mainly obtained from fast-growing (FG) and slow-growing (SG) breeds, and the flavor may vary among them. Xiao et al. reported that the concentration of flavor compounds was higher in native Chinese chickens than in typical commercial broilers [7]. Li et al. found that the volatile compounds in Taihe silky chicken (a black-boned slow-growing chicken) and Cobb chicken were clearly different and identified 24 volatile compounds that can be used as potential biomarkers [8]. However, the differences in the key volatile compounds between SG and FG ducks have not been determined.

An electric nose (E-nose) combined with solid-phase microextraction-gas chromatography-mass spectrometry (SPME-GC-MS) has been adopted for the determination of the flavor compound, which can perform both macro and micro analyses of the meat samples [9,10]. Metabolomics has been widely adopted to assess food quality [11]. Jiang and Bratcher found that metabolomics is an effective method for distinguishing ground beef samples from different sources and backgrounds [12]. The untargeted metabolomics technique was adopted to evaluate the effect of thermal processing on the sturgeon meat flavor [13]. ^1^H nuclear magnetic resonance-based metabolomics analysis explained the variation in precursor flavor substances with age in chicken meat [14]. Collectively, it can be concluded that metabolomics can characterize the metabolites associated with the flavor of food. However, the metabolomic profiles of duck meat and their association with a volatile aroma have not been investigated. Widely targeted metabolomics is a novel metabolomics technique based on multiple reaction monitoring (MRM) using multiple ion monitoring (MIM) survey scans to trigger enhanced product ion (EPI) acquisition, which combines the advantages of the “universality” of untargeted metabolomics and the “precision” of targeted metabolomics [15]. Therefore, the widely targeted metabolomics approach was adopted in this study.

In the present study, SG Liancheng White and FG Cherry Valley ducks were used. The breast muscle samples were dissected, and the volatile compounds were investigated using E-nose combined with SPME-GC-MS analysis. Furthermore, widely targeted metabolomics was performed to investigate the composition of volatile compounds in meat. Our findings provide a more comprehensive knowledge of the key volatile compounds and metabolomic profiles in raw meat and document variations between SG and FG ducks.

## 2. Materials and Methods

### 2.1. Animals Handling

All animal procedures were approved by the Yangzhou Institutional Animal Committee (permit number: YZUDWSY2021-167, 14 September 2021, Jiangsu Province, China). One-day-old female Cherry Valley (SM3 Medium) and Liancheng White (Anas platyrhynchos) ducklings were raised at Zhongke Poultry Co. Ltd. (Suqian, Jiangsu Province, China). During the experiment, the ambient temperature was maintained at 30 °C and gradually decreased to 18 °C to ensure comfort. The indoor humidity was maintained at 65–75%. The ducks were housed on mesh beds, 50 cm above the ground, and fed the same commercial diet (Appendix A).

### 2.2. Slaughtering and Sampling

In the present study, 60 female FG Cherry Valley ducks and 60 female SG Liancheng White ducks were slaughtered at 37 and 65 days of age, respectively. Before their slaughter, the birds were fasted for 12 h, while having free access to water. All the birds were stunned at oscillation frequencies of 900 Hz for 5 s (40 V) followed by exsanguination. The breast muscle samples (*Pectoralis major*) from SG and FG ducks were dissected, cooled, and transported to the laboratory on ice. At 24 h postmortem, the meat samples (20 g for E-nose analysis, 20 g for SPME-GC-MS analysis, and 1 g for widely targeted metabolomics) were collected, immediately frozen in liquid nitrogen, and then stored at −80 °C until a later analysis.

### 2.3. Quantitative Descriptive Sensory Analysis

The quantitative descriptive sensory analysis of the aroma of raw duck meat was carried out according to Katiyo et al. [16]. Briefly, a trained panel of 10 members (5 males and 5 females), aged between 20 and 35 years from Yangzhou university, was recruited. The panel was comprised of people who often dealt with or bought raw duck. The panelists were trained by recognizing and describing the intensity ratings of the standard references (Table 1), and the training sessions were held 4 times for 2 h each time before the sensory analysis. During the sensory analysis, each duck meat sample was labelled with a randomly selected 3-digit code. The researchers set aside a 60 s break between each sample for group members to smell the backs of their hands to neutralize their sense of smell. The results were recorded on a 10-line scale (from no intensity, 0, to a high intensity, 10).

### 2.4. E-Nose Analysis

A portable electronic nose (Airsense, Schwerin, Germany) was used to analyze the volatile compounds in the duck meat. The device consists of 10 different metal-oxide sensors. The sensitivity and selectivity of the sensor for specific volatile compounds are listed in Table 2. The meat sample (5 g) was placed in a 100 mL beaker and was sealed with plastic wrap for 30 min. The headspace suction method was used for the data acquisition. The parameters were set as follows: the sample measurement interval was 1 s; the sensor cleaning time was 80 s; the sample preparation time was 5 s; the internal flow rate was 400 mL min^−1^; and the sample detection time was 80 s. The G/G0 ratio (G0 represents the conductance of the sensors for clean air and G for the sample gas) was used to evaluate the response data of all the sensors.

### 2.5. SPME-GC-MS Analysis

The SPME-GC-MS method, according to Zhan et al., was used to further analyze the volatile compounds in duck meat [17]. Briefly, each sample (2 g) was distributed into 10 mL headspace vials, and 3 mL of the saturated sodium chloride was added. After the rolling, sonication, and sealing with a diaphragm, the vials were heated in a water bath at 80 °C for 30 min. Furthermore, SPME fibers (Supelco, Bellefonte, PA, USA) were exposed to the vial head space at 80 °C for 30 min. A Thermo TRACE DSQ GC-MS system equipped with a TG-5SILMS capillary column (30 m × 0.25 mm, 0.25 μm, Thermo Fisher Scientific, Waltham, MA, USA) was used to perform GC/MS analysis. The fibers were inserted into GC and desorption was completed at 250 °C for 5 min. The temperature started at 50 °C for 3 min, then increased to 250 °C at 6 °C every minute, and finally for 5 min. The parameters were set as follows: the electron energy was 70 eV; the full scan quality range was *m*/*z* 40–650; the inlet temperature was 250 °C; and the ion source temperature was 280 °C. The data were collected using the Chromeleon 7 software (Thermo Fisher Scientific, Waltham, MA, USA). The percentage of the GC peak area is shown as the relative content of each component.

The raw data were processed using the Xcalibur software (version 3.0.63, Thermo Fisher Scientific, Waltham, MA, USA), and the compounds were identified by comparing the mass spectra with those in the NIST 2.0 mass-spectrometry database. The retention index (RI), positive matching values, and negative matching values were used to control the data quality. The compounds with positive or negative matching values greater than 800 (maximum 1000) and RI values lower than 20 were selected. The RI was calculated as follows: RI = 100 × {n + [t(i) − t(n)]/[t(n + 1) − t(n)]}, where n is the number of the carbon atoms, t(i) is the adjusted retention time of the component, t(n) is the adjusted retention time of the n-alkanes with n carbon atoms, and t(n + 1) is the adjusted retention time of n-alkanes with (n + 1) carbon atoms.

### 2.6. Widely Targeted Metabolomics Analysis

#### 2.6.1. Sample Preparation and Extraction

Six samples from each group (FG and SG; 10 individuals mixed in one sample) were prepared for widely targeted metabolomic analysis. The procedures were performed as previously described [14]. Briefly, the meat samples were homogenized with steel balls (30 Hz) for 20 s, and centrifuged (650× *g*, 4 °C) for 30 s. The powder was mixed with 400  μL of 70% methanol–water (internal standard extractant chromatographically pure, BioBioPha/Sigma-Aldrich, St. Louis, MO, USA), oscillated (162× *g*) for 5 min, and centrifuged at 10,390× *g* for 10 min at 4°C. Finally, 300 μL of supernatant was collected and stored at −20 °C for 30 min, and then centrifuged at 10,390× *g* at 4 °C for 3 min. The supernatant was used for further analyses.

#### 2.6.2. UPLC-ESI-QTRAP-MS/MS Conditions and Data Analysis

The samples were analyzed by ultra-performance liquid chromatography-electrospray ionization tandem mass spectrometry (UPLC-ESI-QTRAP-MS/MS, https://sciex.com/, accessed on 14 September 2021). The analytical conditions were as follows: UPLC column was Waters Acquity UPLC HSS T3 C18 (1.8 µm, 2.1 mm × 100 mm, Waters, Milford, MA, USA); the column temperature was 40 °C; the flow rate was 0.4 mL/min; the injection volume was 2 μL; the solvent system was water (0.1% formic acid): acetonitrile (0.1% formic acid); and the gradient program was 95:5 *v*/*v* at 0 min, 10:90 *v*/*v* at 11.0 min, 10:90 *v*/*v* at 12.0 min, 95:5 *v*/*v* at 12.1 min, and 95:5 *v*/*v* at 14.0 min. Triple quadrupole scans and linear ion trap were obtained on a triple quadrupole-linear ion trap mass spectrometer (QTRAP), which was controlled by Analyst 1.6.3 software (AB SCIEX, Framingham, MA, USA). The ESI source operation parameters were as follows: the source temperature was 500 °C; the ion spray voltage was 5500 V (positive) and −4500 V (negative); the ion source gas I, gas II, and the curtain gas were set at 55, 60, and 25.0 psi, respectively; and the collision gas was high. Instrument tuning and mass calibration were performed using 10 and 100 μmol/L polypropylene glycol solutions in triple quadrupole scans and Linear ion trap modes, respectively. A specific set of multiple reaction monitoring (MRM) transitions was monitored based on the metabolites eluted at each time period. The metabolite identification was performed using the self-built MWDB database (Metware Biotechnology Co., Ltd. Wuhan, China) and other public databases (HMDB, MassBank, METLIN, and MoTo DB). The peak area integration was performed on the obtained metabolite mass spectral peaks, and the mass spectral peaks of the metabolites in the different samples were integrated. In addition, a quality control (QC) sample was prepared by mixing the sample extracts and was used to monitor the repeatability of the analysis samples under the same treatment. During the instrumental analysis, one quality control sample was inserted into every 10 test samples to monitor the repeatability of the analysis process (Appendix A).

### 2.7. Statistical and Bioinformatic Analysis

For quantitative descriptive sensory, E-nose, and SPME-GC-MS analyses, the statistical software SPSS (version 25.0, SPSS, Chicago IL, USA) was used to compare the repeated measurements by a t-test between the SG and FG groups. *p* ≤ 0.05 was considered to be statistically significant. All the data are presented as the means ± standard error (SE).

For widely targeted metabolomics analysis, the multivariate statistical analyses of unsupervised principal component analysis (PCA) and hierarchical cluster analysis (HCA) were performed using R software (www.r-project.org, accessed on 21 September 2021). Supervised multiple regression orthogonal partial least squares discriminant analysis (OPLS-DA) was performed using R package MetaboAnalystR. The differential metabolites between the groups were determined by combining the fold change and threshold variables in the projection (VIP) value of the OPLS-DA model; a VIP value ≥1 and a fold change ≥ 2 (up-regulated) or ≤ 0.5 (down-regulated) were set. The identified differential metabolites were annotated using the Kyoto Encyclopedia (KEGG) and Genomes database (http://www.kegg.jp/kegg/compound/, accessed on 25 September 2021). Pearson’s correlation was used to analyze the relationships between the different metabolites in the raw duck meat.

## 3. Results

### 3.1. Quantitative Descriptive Sensory and E-Nose Analysis of Duck Meat

Quantitative descriptive sensory and E-nose analysis was performed to obtain comprehensive information about the volatile compounds in SG and FG duck meat. The intensity of the fishy note was significantly higher in SG meat than in FG meat (Table 3). Sensors W1S, W1W, W2S, W2W, and W5S displayed higher values than the other sensors in both breeds, indicating that the volatile components were similar in the FG and SG meat (Figure 1A). Among the top five sensitive sensors, the response value of W1S was the highest in the two breeds (Table 4), and it was significantly higher in the SG meat than in FG meat samples (*p* < 0.05). These results indicate that methane, inorganic sulfides, alcohol, sulfur organic compounds, and nitrogen oxides are the main volatile components of duck meat. SG meat had a higher methane content than FG meat. Linear discriminant analysis (LDA) was also performed (Figure 1B), which could efficiently separate the SG and FG meat. Overall, the SG group had a higher fishy note and duck odors and contained more volatile compounds than the FG group. Methane was the main volatile compound in both breeds and was higher in the SG meat than in FG meat.

### 3.2. SPME-GC-MS Analysis of Duck Meat

To better understand the contribution of the volatile compounds from SG and FG duck meat, SPME-GC-MS analysis was performed. As shown in Table 5, the major volatile compounds in the raw duck meat were hexanal, nonanal, octanal, heptanal, and 2-pentylfuran, which accounted for over 60% of the contribution rates in both duck breeds. Nonanal, octanal, and octenal accounted for a higher contribution rate in SG meat than in FG meat, while octenol had advantages in the FG meat. Taken together, hexanal, nonanal, octanal, heptanal, and 2-pentylfuran were confirmed as the key odorants in raw duck meat, and octenol was a key volatile compound in FG meat, but not in SG meat.

### 3.3. Metabolomic Profiles of Duck Meat

Widely targeted metabolomics were used to determine the overall profile of the metabolites in SG and FG breast meat. A total of 833 metabolites were identified and annotated (Figure 2A), which could be divided into 16 different categories, including 218 amino acids and their metabolomics, 109 organic acids and their derivatives, 94 glycerophospholipids, 84 fatty acyls, 73 nucleotides and their metabolomics, 55 carboxylic acids and their derivatives, 47 benzene and substituted derivatives, 42 heterocyclic compounds, 41 alcohols and amines, 19 aldehydes, ketones and esters, 14 coenzymes and vitamins, and others. An unsupervised PCA was performed to evaluate the differences between the samples (Figure 2B). The results showed that the first two principal components explained 43.82% of the total variation (PC1 = 31.27% and PC2 = 12.55%). Six biological replicates from each group were clustered together, and the SG and FG meat samples were clearly separated. Meanwhile, the metabolomic profiles of the duck meat samples were visualized using a heat map (Figure 2C). Taken together, the metabolomics of the duck meat were mainly amino acids, organic acids, glycerophospholipids, fatty acyls, nucleotides, and carboxylic acids. Unsupervised PCA separated the SG and FG meat samples.

### 3.4. Identification of Differential Metabolites between SG and FG Meat

To identify the differential metabolites between the SG and FG meat, OPLS-DA using supervised pattern recognition was adopted. Pairwise comparisons of SG and FG using the OPLS-DA model are shown in Figure 3A. In our model, the R^2^Y and Q^2^ scores (prediction parameters) were 0.997 and 0.93, respectively, indicating that the model was feasible and efficient. Clear separations were observed between the SG and FG meat in the OPLS-DA score plot (Figure 3B).

Metabolites with a fold change ≥2 (up-regulated) or ≤0.5 (down-regulated), a *p*-value ≤0.05, and a VIP ≥1 were regarded as significant differential metabolites between SG and FG meat (Figure 3C and Appendix A). In total, 233 differential metabolites were identified. Among them, 121 metabolites were down-regulated and 112 metabolites were up-regulated in the FG group compared with those in the SG group. Nine fatty acyls were among the top 10 up-regulated metabolites, including carnitine (C14:1-OH, C14:1, C18-OH, C16:1, C8:1, and others) and hexadecanedioic acid. The levels of carnitine, glycine, meso-erythritol, 2-aminobutyric acid, trigonelline, eicosapentaenoic acid, pyridoxine, and 5-aminovaleric acid were also up-regulated in the FG group compared to those in the SG group.

Eight types of sugars, belonging to carbohydrates and their metabolites, were among the top 10 down-regulated metabolites, including lactulose, lactose, and maltotriose. The glucose and glucose-6-phosphate levels were also down-regulated. In addition, several amino acids and their metabolomics (such as proline, tryptophan, and securinine) were lower in the FG group than in the SG group. In addition, the trimethylamine N-oxide levels were higher in the SG group than in the FG group. The differential metabolites between FG and SG meat were visualized and classified using HCA (Figure 3D).

Overall, fatty acyls, glycine, 2-aminobutyric acid, trigonelline, eicosapentaenoic acid, and others were higher in the FG group, while the sugars, amino acids, trimethylamine N-oxide, and others were higher in the SG group.

### 3.5. KEGG Enrichment and Pearson Correlation Analysis of Differential Metabolites between SG and FG Meat

The biological pathways associated with the identified differential metabolites were identified using KEGG pathway analysis (Figure 4A). The enriched pathway terms were primarily carbohydrate digestion and absorption, the starch and sucrose metabolism, the nicotinate and nicotinamide metabolism, the amino and nucleotide sugar metabolism, and taste transduction. A chord diagram was constructed to analyze the correlation between the metabolites (Figure 4B). Most metabolites belonging to the carbohydrates or fatty acyls showed the same trend, and sugar and carnitine were negatively correlated with each other. In addition, trimethylamine N-oxide was highly correlated with indole-3-acetic acid, dihydroxybenzoic acid, and urea.

## 4. Discussion

Unlike cooked meat, raw meat has little smell and only a slight serum-like taste, described as metallic, salty, and ‘bloody’ with a sweet aroma [18]. However, the volatile compounds released in raw meat are still important factors that affect consumers’ purchase decisions. A few consumers consider that duck meat has a distorted smell (fishy/off-odour), which is not preferred [19]. In addition, the duck meat from SG and FG broilers may have their own aromas. Neither the macro and micro of aroma, nor the formation of volatile compounds in duck meat, is clear.

In the present study, we first conducted an E-nose analysis of SG and FG duck meat. The use of electronic noses in the food industry is attractive because of the low cost of the instruments and the good efficiency [20]. We found that W1S, W1W, W2S, W2W, and W5S displayed higher values than the rest of the sensors in both breeds, indicating that methane, inorganic sulfides, alcohol, sulfur organic compounds, and nitrogen oxides were the main aroma components of duck meat, and SG and FG duck meat samples had similar odor characteristics. Dong et al. also found that these five sensors accounted for the majority of egg yolks and albumen [21]. Furthermore, the SG duck can be distinguished from the FG duck by LDA analysis, and the SG duck had a higher odor than the FG duck, especially in methane. Our results are similar to those of Chen et al. who used E-nose to differentiate Beijing-you chickens (slow-growing) from Arbor Acres broilers (fast-growing) [22]. Li et al. also found that different goat breeds had different E-nose sensor response values [23]. Therefore, E-nose could be an effective tool for distinguishing between SG and FG duck meat.

Next, to identify the specific volatile compounds and explain their contribution to the aroma of raw duck meat, SPME-GC-MS analysis was performed. Hexanal, nonanal, octanal, heptanal, and 2-pentylfuran were the major compounds, accounting for over 60% of the contribution rates in both breeds. Hexanal accounted for 39.29% and 40.57% of the volatile compounds in SG and FG ducks, respectively. Hexanal is the main product of linoleic acid oxidation via 13-hydroperoxide and is used as an indicator of the lipid oxidation processes [24]. The smell of hexanal is described as grass, tallow, and fatty, and it has been characterized as a major volatile compound in mutton, pork, and chicken [8,9,25]. Additionally, Liu et al. suggested that saturated linear aldehydes, such as hexanal, octanal, and nonanal produce pungent, irritating, and unpleasant odors, which have been proven to be the main fishy substances [26]. 2-pentylfuran (green bean and butter note) is also known to be responsible for the off-flavor note and reversion flavor in raw meat [27]. In our study, nonanal and octanal accounted for a higher contribution rate in SG than in FG meat, which indicated that the aroma of SG meat may be more fatty and fishy. Furthermore, FG meat had advantages in octenol compared with SG meat, with the corresponding values of 5.77% and 0.45%, respectively. Octenol is known as “mushroom alcohol,” with green and mushroom notes [28]. It has also been characterized in different types of meats. Taken together, we conclude that saturated linear aldehydes and 2-pentylfuran give duck meat a fatty smell. SG meat may be fishier because of higher rates of nonanal and octanal, while FG meat has a greener and mushroom note because of a higher rate of octenol.

Finally, a widely targeted metabolomic approach was used to investigate the key metabolites in duck meat and identify some flavor-associated compounds. In the last two decades, metabolomics has been widely employed and is expected to increase the knowledge and enrich the methodologies for the prediction of meat quality [29,30]. Of the current techniques, widely targeted metabolomic analysis is targeted at a large scale and can be compared. However, some isomers that could not be distinguished still existed. In the present study, a total of 833 metabolites were identified and annotated; the metabolomics of the duck meat were mainly amino acids, organic acids, glycerophospholipids, fatty acyls, nucleotides, and carboxylic acids. A previous study on ducks detected some of these metabolites [31]. OPLS-DA analysis was performed to identify the differential metabolites between SG and FG ducks. Carnitines, which are fatty acyls, were among the top up-regulated metabolites in FG compared to SG meat. Carnitine plays a key role in the lipid metabolism and β-oxidation. It is used to transport long-chain fatty acids to the mitochondria, where they are oxidized to produce energy. Several experiments have confirmed that a dietary supplementation with carnitine in animals can inhibit a fat accumulation and decrease the fat content [32,33]. The lower lipid oxidation in FG duck meat may be attributed to its higher carnitine content. In addition, some potential biomarkers for foods, such as 2-aminobutyric acid and 2-aminoadipic acid, and coenzymes and vitamins, such as trigonelline and pyridoxine, were also found to be higher in FG meat.

Eight types of sugars, belonging to carbohydrates and their metabolites (including lactulose, lactose, glucose, and glucose-6-phosphate), were down-regulated in the FG group compared to the SG group. It has been found that some sugars, which included glucose, fructose, and ribose may contribute to the sweet flavor in meat [34]. In addition, glucose and glucose 6-phosphate can increase the “meaty” and “roasted” aroma [35]. In addition, some amino acids, such as methionine, tryptophan, and proline, were higher in SG meat. Methionine and tryptophan are essential amino acids that the body cannot synthesize and must be obtained from the diet. It is required for proper growth and the development of humans, other mammals, and birds [36]. Therefore, SG duck meat may have advantages in cooking and roasting because of its higher sugar and amino acid contents. However, trimethylamine N-oxide was also up-regulated in SG ducks. Trimethylamine N-oxide is an oxidation product of trimethylamine that can be derived from dietary carnitine or lecithin [37]. Trimethylamine N-oxide does not have any odor or taste. However, it can also decompose into trimethylamine. The odor of trimethylamine is described as “old-fishy” or “fish house-like”. Trimethylamine has an ammoniacal odor in its pure state; it is not responsible for the fishy odor alone, but its reaction with fat in the tissue creates a fishy odor [38]. Additionally, trimethylamine N-oxide was highly correlated with indole-3-acetic acid, dihydroxybenzoic acid, and urea in duck meat. Urea has no flavors, but it can be quickly decomposed into ammonia and CO_2_ by bacterial urease. The odor of ammonia is pungent and leads to an unacceptable meat quality [39]. Therefore, the storage and preservation of duck meat should be monitored carefully. Taken together, higher sugar and amino acid contents led to a sweet and meaty aroma, while a higher trimethylamine N-oxide content may generate fishy notes in SG ducks.

In the present study, we characterized the volatile compounds and key metabolites related to the aroma of SG and FG duck meat samples (Figure 5). These results will be useful for increasing the recognition and consumption of duck meat. E-nose and SPME-GC-MS analysis can be effective tools to distinguish duck meat from different breeds. Widely targeted metabolomics provides a basis for understanding the molecular mechanism of aroma formation in duck meat. Next, we will verify the key metabolites and their use in duck breeding.

## 5. Conclusions

In summary, E-nose analysis combined with SPME-GC-MS analysis showed a general uniformity in the main aroma of SG and FG ducks. Methane, inorganic sulfides, alcohol, sulfur organic compounds, and nitrogen oxides are the macro aroma components, while hexanal, nonanal, octanal, heptanal, and 2-pentylfuran are the micro volatile compounds in duck meat. SG duck had a higher odor than the FG duck meat, especially in regarding the methane compound. SG duck meat may be fatty and fruity because of the higher rates of nonanal and octanal, while FG meat has a greener and mushroom note because of its higher rate of octenol. The widely targeted metabolomics showed that the lower lipid oxidation products in FG duck meat may be attributed to the higher carnitine content. Higher sugar and amino acid contents led to a sweet and meaty aroma, while a higher trimethylamine N-oxide content may generate fishy notes in SG ducks. Our study increases the understandings of the aroma of raw duck meat and the basic molecular mechanism of the formation of an aroma, which is helpful for duck breeding.

## Figures and Tables

**Figure 1 foods-11-03975-f001:**
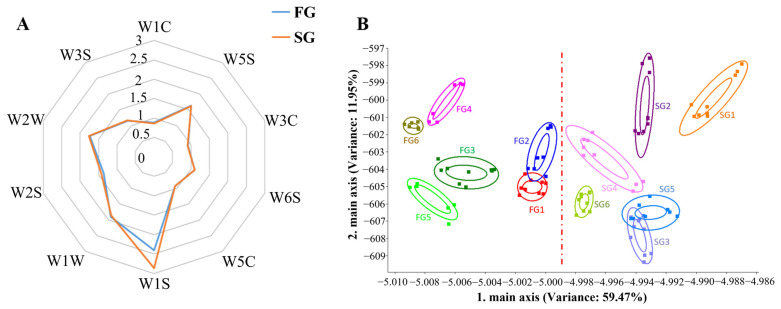
Electronic nose analysis of slow-growing and fast-growing (SG and FG) duck breast meat. (**A**) Radar chart of the E-nose in SG and FG meat. (**B**) The linear discriminant analysis (LDA) of the E-nose in SG and FG meat.

**Figure 2 foods-11-03975-f002:**
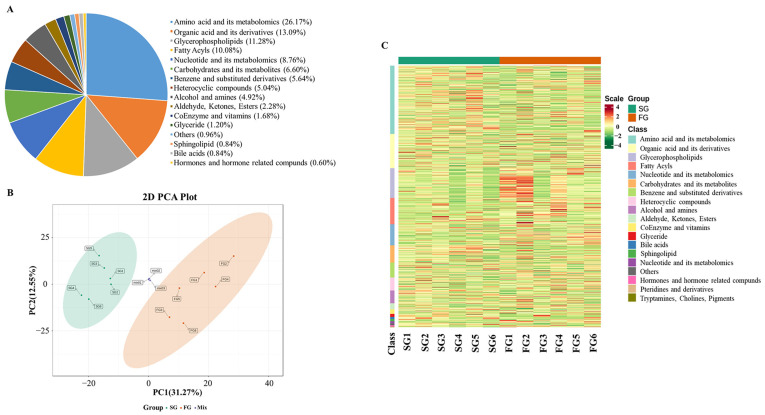
Overall profile of metabolites in slow-growing and fast-growing (SG and FG) duck breast meat. (**A**) The 833 metabolites divided into 16 different categories of meat samples. (**B**) Unsupervised principal component analysis (PCA) of two breeds with six biological replicates. (**C**) Heat map revealing expression pattern of metabolites among SG and FG meat.

**Figure 3 foods-11-03975-f003:**
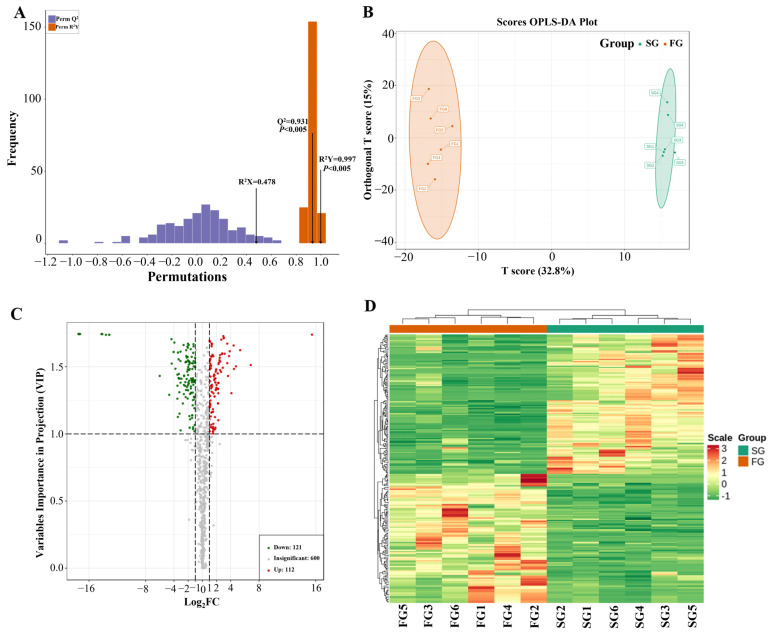
Identification of differential metabolites between slow-growing and fast-growing (SG and FG) meat. (**A**) Model validation of supervised multiple regression orthogonal partial least-squares discriminant analysis (OPLS-DA) pairwise comparisons. (**B**) The score plots of OPLS-DA pairwise comparisons. (**C**) Volcano plots of differential metabolites. (**D**) Hierarchical clustering analysis (HCA) of the metabolites.

**Figure 4 foods-11-03975-f004:**
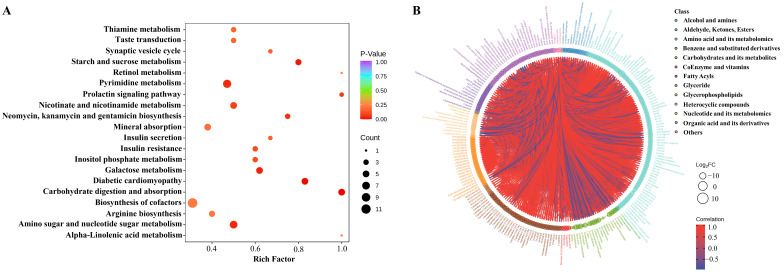
KEGG enrichment and Pearson correlation analysis of differential metabolites between SG and FG meat. (**A**) KEGG annotation and enrichment analysis of differential metabolites. (**B**) The chord diagram of differential metabolites by Pearson correlation analysis.

**Figure 5 foods-11-03975-f005:**
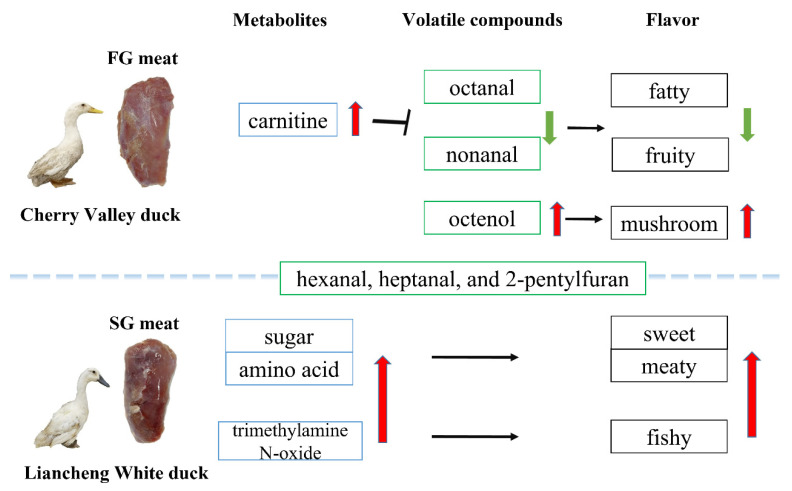
Schematic illustrating key volatile aroma compounds in SG and FG raw duck meat. Hexanal, heptanal, and 2-pentylfuran were the common volatile compounds in duck meat. Higher carnitine was present in FG meat, which reduce lipid deposition to produce more octanal and nonanal. SG meat may be fatty and fruity for higher rate of nonanal and octanal. FG meat has a more mushroom note for higher rate of octenol. Higher sugar and amino acid contents led to sweet and meaty aroma, while higher trimethylamine N-oxide content may generate fishy note in SG meat.

**Table 1 foods-11-03975-t001:** Quantitative descriptive sensory analysis of the aroma of raw duck meat.

Descriptor	Definition	Rating Scale	Standard Reference (Extreme Level)
Bloody	Aromatic associated with raw lean meat, blood, serum or metal/iron	0 = no bloody odor 10 = intense bloody odor	Raw beef liver
Pungent	Very strong, sharp smell	0 = no pungent odor 10 = intense pungent odor	Reminiscent of a rotten freshwater fish
Fishy	Aromatic associated with spoiled fish	0 = no fishy odor 10 = intense fishy odor	Reminiscent of a rotten freshwater fish
Rotten egg	Aromatic associated with rotten eggs	0 = no rotten egg odor 10 = intense rotten egg odor	Reminiscent of a rotten egg
Ammonia-like	Aromatic associated with ammonia	0 = no ammonia-like odor 10 = intense ammonia-like odor	0.2% ammonia in water

**Table 2 foods-11-03975-t002:** Sensors used and their performance descriptions in PEN3 electronic nose.

Sensors	Sensitive Characteristics	Reference Volatile Compounds
W1C	Aromatic compounds	Methylbenzene, 10 ppm
W5S	Nitrogen oxides	NO_2_, 1 ppm
W3C	Ammonia and aromatic compounds	Benzene, 10 ppm
W6S	Hydrogen	H_2_, 100 ppb
W5C	Hydrocarbons, aromatic compounds	Propane, 1 ppb
W1S	Methane, broad range	CH_3_, 100 ppm
W1W	Inorganic sulfides	H_2_S, 1 ppm
W2S	Alcohols, aldehydes and ketones, broad range	CO, 100 ppm
W2W	Aromatic compounds, sulfur organic compounds	H_2_S, 1 ppm
W3S	Long-cyclic alkane	CH_3_, 100 ppm

**Table 3 foods-11-03975-t003:** Comparison on descriptive sensory intensity from slow- and fast-growing duck meat.

Descriptor	SG	FG	*p*-Values
Bloody	1.50 ± 0.17	1.30 ± 0.15	0.388
Pungent	1.40 ± 0.16	1.30 ± 0.15	0.660
Fishy	2.60 ± 0.22 ^a^	1.50 ± 0.22 ^b^	0.003
Rotten egg	1.20 ± 0.13	1.30 ± 0.15	0.628
Ammonia-like	1.70 ± 0.26	1.60 ± 0.16	0.749

Note: ^a,b^ means ± SE (*n* = 6) with different superscript are significantly different in the same line (*p* < 0.05).

**Table 4 foods-11-03975-t004:** Comparison on typical response values of E-nose sensors (G/G0 ratio) from slow- and fast-growing duck meat.

Sensors	SG	FG	*p*-Values
W1C	0.86 ± 0.01 ^b^	0.88 ± 0.01 ^a^	0.018
W5S	1.62 ± 0.10	1.63 ± 0.11	0.878
W3C	0.91 ± 0.01 ^b^	0.92 ± 0.01 ^a^	0.024
W6S	1.10 ± 0.01	1.10 ± 0.00	0.078
W5C	0.92 ± 0.01	0.93 ± 0.01	0.112
W1S	2.86 ± 0.37 ^a^	2.40 ± 0.33 ^b^	0.045
W1W	1.87 ± 0.06	1.90 ± 0.14	0.567
W2S	1.43 ± 0.05 ^a^	1.36 ± 0.05 ^b^	0.048
W2W	1.76 ± 0.07	1.74 ± 0.09	0.698
W3S	1.17 ± 0.01 ^a^	1.15 ± 0.00 ^b^	0.000

Note: ^a,b^ means ± SE (*n* = 6) with different superscript are significantly different in the same line (*p* < 0.05).

**Table 5 foods-11-03975-t005:** Comparison on contribution rates (%) of volatile compounds from slow- and fast-growing duck meat.

Compounds	Slow-Growing	Fast-Growing	Flavors and Aromas
Hexanal	39.29 ± 2.95	40.57 ± 2.30	Green, grass
Nonanal	12.65 ± 0.89 ^a^	8.81 ± 1.19 ^b^	Green, stale
Octanal	7.03 ± 0.34 ^a^	4.39 ± 0.61 ^b^	Fruity, sour
Furan, 2-pentyl-	4.21 ± 0.87	4.18 ± 0.62	Metallic, green, earthy, beany
Heptanal	3.64 ± 0.15	3.11 ± 0.27	Cheesy, fatty
Octenol	0.45 ± 0.45 ^b^	5.77 ± 1.89 ^a^	Mushroom
Octenal	2.51 ± 0.37 ^a^	1.29 ± 0.43 ^b^	Fatty
Octanedione	1.78 ± 0.58	3.88 ± 1.07	Warmed over flavor, lipid oxidation
Pentanal	1.59 ± 0.19	1.46 ± 0.25	Pungent
Formic acid, octyl ester	0.44 ± 0.23	0.11 ± 0.10	-
Heptanol	0.49 ± 0.16	0.23 ± 0.10	Green
Heptenal	0.77 ± 0.31	0.38 ± 0.16	Fatty
Pentanol	0.45 ± 0.08 ^b^	0.8 ± 0.14 ^a^	Fruity, grassy, sweet
Benzaldehyde	0.79 ± 0.11	1.5 ± 0.39	Almond
Carbamodithioic acid, diethyl-, methylester	0.59 ± 0.13	0.63 ± 0.31	-
Propanal, 3-(methylthio)-	0.26 ± 0.08	0.33 ± 0.10	-
Octanol	1.04 ± 0.27	0.94 ± 0.28	Citrus
Decadienal	0.57 ± 0.22	0.26 ± 0.15	Fatty, meat
Tridecenal	1.52 ± 0.73	0.12 ± 0.08	Waxy, citrus rind, fatty, soapy
Decanal	0.28 ± 0.09	0.27 ± 0.12	Flowery
Cyclohexenol	0.42 ± 0.28	0.74 ± 0.45	Caramelized, phenolic, and floral aroma
Heptanone	0.27 ± 0.11 ^b^	0.98 ± 0.22 ^a^	Musty, spicy
Octenol	0.04 ± 0.04 ^b^	0.7 ± 0.26 ^a^	Green apple
Pentadecanal-	0.19 ± 0.09	0.24 ± 0.14	Citrus rind and fresh waxy odors
Nonenal	0.65 ± 0.25	0.39 ± 0.20	Fishy, tallow
Hexanol	0.1 ± 0.06	0.23 ± 0.10	Green
Nonadienal	0.37 ± 0.18	-	Fishy, waxy

Note: ^a,b^ means ± SE (*n* = 6) with different superscript are significantly different in the same line (*p* < 0.05).

## Data Availability

Data is contained within the article or Appendix A.

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
