# Peer review of "Comparative Characterization of Key Volatile Compounds in Slow- and Fast-Growing Duck Raw Meat Based on Widely Targeted Metabolomics"

_foods, 2022, doi:10.3390/foods11243975_

Round 1
Reviewer 1 Report (Previous Reviewer 3)
The authors have addressed the initial concerns of the reviewers and the manuscript is now in excellent condition. There are just a couple of minor issues that should be addressed.
Line 30 I’d prefer “high nutritional value”
Line 154 should read “2.1 mm x 100 mm” use the common multiplication term “x” instead of “*”.
Throughout the document L-2-amino-butyric acid (and a couple of other amino acids) is used, the use of metabolomics does not necessarily detect the difference between L- and D- isoforms and these should be removed. This would also ensure more consistency throughout the document with other compounds listed.
Similarly, 1-octen-3-ol suggests a level of analytical certainty about the compound, common names such as octenol may be more appropriate. This should be applied to all chemical names to be consistent throughout the document. If there is absolute certainty with analytical standards then the precise detail is justified. The authors should consider if the analytical techniques they have used, with retention index matched to NIST library data are accurate enough to categorically ensure the compounds detected are the isomers they are claimed to be.
Care needs to be taken with specialist terms including “P value” (Table 3 and line 264) which should be written p-values. Hopefully all of these will be picked up during the typesetting process.
Author Response
Response to Reviewer 1 Comments
Point 1: The authors have addressed the initial concerns of the reviewers and the manuscript is now in excellent condition. There are just a couple of minor issues that should be addressed.
Response 1: Thank you again for your reviews and great efforts concerning our manuscript. We have revised the manuscript according to the comments. Revised portion are highlight in yellow in the revised manuscript.
Point 2: Line 30 I’d prefer “high nutritional value”.
Response 2: We have replaced “nutritive value” with “nutritional value” (Line 30-31).
Point 3: Line 154 should read “2.1 mm x 100 mm” use the common multiplication term “x” instead of “*”.
Response 3: We have replaced “*” with “x” (Line 154).
Point 4: Throughout the document L-2-amino-butyric acid (and a couple of other amino acids) is used, the use of metabolomics does not necessarily detect the difference between L- and D- isoforms and these should be removed. This would also ensure more consistency throughout the document with other compounds listed.
Response 4: Thanks for the advice. Indeed, the use of metabolomics does not necessarily detect the difference between L- and D- isoforms. We have removed “L-” and checked isomers throughout the document (Line 270, 271, 276, 280, 360, 368).
Point 5: Similarly, 1-octen-3-ol suggests a level of analytical certainty about the compound, common names such as octenol may be more appropriate. This should be applied to all chemical names to be consistent throughout the document. If there is absolute certainty with analytical standards then the precise detail is justified. The authors should consider if the analytical techniques they have used, with retention index matched to NIST library data are accurate enough to categorically ensure the compounds detected are the isomers they are claimed to be.
Response 5: Thanks for the advice. For SPME-GC-MS analysis, with retention index matched to NIST library data, it still could not detect the differences between isomers. Thus, the “octenol” is more appropriate than “1-octen-3-ol”. We have checked the isomers and corrected chemical names throughout the document. (Line 19, 228-230, Table 5, 337-342 and Figure 5)
Point 6: Care needs to be taken with specialist terms including “P value” (Table 3 and line 264) which should be written p-values. Hopefully all of these will be picked up during the typesetting process.
Response 6: We have replaced “P value” with “p-values” (Table 3, 4 and line 264). We have checked and revised such mistakes throughout the manuscript.
Reviewer 2 Report (Previous Reviewer 2)
I appreciate the great efforts that the authors have made in response to my comments. The revision clarifies all the points I raised. In my opinion, it helps understand better the current manuscript.
Author Response
Response to Reviewer 2 Comments
Point 1: I appreciate the great efforts that the authors have made in response to my comments. The revision clarifies all the points I raised. In my opinion, it helps understand better the current manuscript.
Response 1: Thank you again for your great efforts concerning our manuscript. Those comments are all valuable and very helpful for revising and improving our paper.
Reviewer 3 Report (Previous Reviewer 1)
The observations were attended
Author Response
Response to Reviewer 3 Comments
Point 1: The observations were attended.
Response 1: Thank you again for your kind reviews and great efforts concerning our manuscript.
This manuscript is a resubmission of an earlier submission. The following is a list of the peer review reports and author responses from that submission.
Round 1
Reviewer 1 Report
Title: Comparative characterization of key volatile compounds in slow- and fast-growing duck raw meat based on widely targeted metabolomics.
The manuscript presents information of interest to the poultry meat industry (mainly in China) and to the scientific community. The analytical techniques used allowed to characterize and differentiate the volatile compounds present in duck meat of two breeds.
The document is well written, the results were properly interpreted and easy to understand. However, there are some minor observations:
Line 115: 2.5 section instead 2.6 section.
Lines-117-124: In this paragraph, Centrifugation units change to g instead of rpm.
Line 138: P ≤ 0.05 were considered statistically significant (Repeated sentence).
Line 241 and 253: Pearson correlation instead personal correlation.
Reviewer 2 Report
The writing is quite interesting, however, a correlation of the physicochemical variables with a sensory panel evaluation would have been pertinent given the relevance of the study regarding characterization of key volatile compounds (descriptive sensory analysis or at least consumer sensory testing). Can be perceived differences in sensory attributes when duck meat is cooked?
Reviewer 3 Report
e-nose and metabolomic analysis of meat from fast and slow growing ducks
This paper investigates the volatile and non-volatile compounds that may contribute to purchasing decisions of raw duck meat. As the meat has not been cooked prior to analysis it is not possible to make comments on the flavour compounds associated with taste. The research undertaken is valid and is mostly conducted well. Data tables, figures and supplementary data are presented well.
The paper is let down by lack of editing and use of grammar, making the paper difficult to read and leading to confusion of the points that the authors are trying to make. I strongly recommend this paper edited for use of language, grammar and spelling, then scientifically edited prior to resubmission. There is a significant lack of attention to detail that is required.
Major concern -
It is overstated that a “widely target metabolomics” approach was taken, there are a couple of issues with this.
Firstly, the methods describe a targeted metabolomics MRM approach and not a “widely targeted” approach which uses a targeted and a scanning approach. The authors need to write out the metabolomics methods in full. The authors should also describe what they mean by a widely targeted metabolomics in the introduction as this is not common knowledge for the target audience.
Secondly, on writing out the methods for the metabolomics, consideration should be given to the chromatography. What compounds will be separated on the column? Is separating and detecting sugars, amino acids, carboxylic acids and lipids all on the same column possible? Have any standards or quality control been run to confirm any of these compounds?
Thirdly, in the supplementary table there are multiple entries for the same compound. This list needs to be reviewed and edited and the results in the manuscript adjusted accordingly. This includes several phospholipids, fatty acids, “sugars”, amino acids, it appears that the software is giving multiple annotations for the same peak. There are also a lot of dipeptides and tripeptides that might be considered differently to amino acids and need more evidence to confirm their presence.
For the SPME analysis, it should be stated what software was used to analyse the data and include quality controls to confirm compounds identified. What units are used for both 2 and 3?
The legends and writing on the figures is too small to read.
Some minor points
Line 28 Duck is the major source
Line 30 what is good quantities? High quantities? Within the daily limit?
Line 44 -declared is a strong word
Line 52 – it should be “SPME”, throughout the paper.
Line 59 – “metabolites associated with”
Line 67 – “our findings provide a”
Line 92 – should it be “reference volatile compounds”?
Line 101 – Title case for names. – Zhan et a.
Line 102 – remove concrete
Line 106 – Thermo Trace DSQ? GCMS
Line 117 – were prepared for metabolomics analysis.
Line 126 – chromatography not chromatoid
Line 130 – what is Pet.?
Line 126 – need to give details of mass spec source conditions and chromatography.
Line 130 – what is the specific set of MRM and what are the time periods. Were controls used to verify each compound?
Line 145 – sentences need editing
Line 147 – why do metabolites need to be annotated with KEGG when a targeted metabolomics method has been used? Or is this for pathway analysis?
Line 157 – what is broad methane and broad alcohol? Correct terminology should be used.